# Data-Driven Lexical Normalization for Medical Social Media †

**Anne Dirkson** [1,*] **, Suzan Verberne** [1] **, Abeed Sarker** [2] **and Wessel Kraaij** [1]

1    Leiden Institute for Advanced Computer Science, Leiden University, 2333 CA Leiden, The Netherlands
2    Department of Biomedical Informatics, Emory University, Atlanta, GA 30322, USA
*    Correspondence: a.r.dirkson@liacs.leidenuniv.nl
†    This paper is an extended version of our paper published in Social Media Mining for Health Applications workshop (SMM4H), ACL 2019.

**Abstract:** In the medical domain, user-generated social media text is increasingly used as a valuable complementary knowledge source to scientific medical literature. The extraction of this knowledge is complicated by colloquial language use and misspellings. However, lexical normalization of such data has not been addressed effectively. This paper presents a data-driven lexical normalization pipeline with a novel spelling correction module for medical social media. Our method significantly outperforms state-of-the-art spelling correction methods and can detect mistakes with an $F_1$ of 0.63 despite extreme imbalance in the data. We also present the first corpus for spelling mistake detection and correction in a medical patient forum.

**Keywords:** spelling correction; social media; health; natural language processing; lexical normalization

## 1. Introduction

In recent years, user-generated data from social media that contains information about health, such as patient forum posts or health-related tweets, has been used extensively for medical text mining and information retrieval (IR) [1]. This user-generated data encapsulate a vast amount of knowledge, which has been used for a range of health-related applications, such as the tracking of public health trends [2] and the detection of adverse drug responses [3]. However, the extraction of this knowledge is complicated by non-standard and colloquial language use, typographical errors, phonetic substitutions, and misspellings [4–6]. This general noisiness of social media text is only aggravated by the complex medical domain [1].

The noisiness of medical social media can be reduced through lexical normalization: the transformation of non-standard text to a standardized vocabulary. Nonetheless, lexical normalization for medical social media has not been explored thoroughly. Medical lexical normalization methods (i.e., abbreviation expansion [7] and spelling correction [8,9]) have mostly been developed for clinical records or notes. Although clinical records also contain many domain-specific abbreviations and misspellings, their contents are typically focused solely on the medical domain. In contrast, social media text typically covers a wider vocabulary including colloquial language and layman's terms for medical concepts [1,10]. For medical social media, some recent studies have explored the related task of concept normalization (i.e., the mapping of tokens to standardized concept IDs in an ontology) [1]. (For example, lexical normalization of "pounding hed" would output "pounding head", whereas concept normalization would aim to map it to the concept of Headache in a medical ontology such as SNOMED CT. A major difference between lexical and concept normalization is that the latter is constrained to terms of a pre-defined category (e.g., symptoms), whereas lexical normalization is unconstrained and can include any term.) Community-driven research on the topic

has been boosted by the public release of relevant annotated datasets (CSIRO Adverse Drug Event Corpus (Cadec) [11], Psychiatric Treatment Adverse Reactions data set (psyTAR) [12] and the shared tasks of the Social Media Mining for Health Applications (SMM4H) workshop [13,14]). However, these available annotated datasets for concept normalization do not annotate misspellings explicitly and are thus not suitable for evaluating lexical normalization. In fact, as of yet, there are no publicly available annotated datasets for lexical normalization in medical social media.

Currently, the most comprehensive benchmark for lexical normalization in *general-domain* social media is the ACL W-NUT 2015 shared task (https://noisy-text.github.io/norm-shared-task.html) [15]. The current state-of-the-art system for this task is MoNoise [16]. However, this system is supervised and uses a lookup list of all replacement pairs in the training data as one of its important features. The training data from the task consist of 2950 tweets with a total of 3928 annotated non-standard words [15]. As extensive training data are unavailable for medical social media, such supervised systems cannot be employed in this domain. The best unsupervised system available is a modular pipeline with a hybrid approach to spelling, developed by Sarker [5]. Their pipeline also includes a customizable back-end module for domain-specific normalization. However, this back-end module relies on: (i) a standard dictionary supplemented manually with domain-specific terms to detect mistakes; and (ii) a language model of distributed word representations (word2vec) of generic Twitter data to correct these mistakes (for more details, see Section 3.2.2). For domains that have many out-of-vocabulary (OOV) terms compared to the available dictionaries and language models, such as medical social media, this is problematic.

Manual creation of specialized dictionaries is an unfeasible alternative: medical social media can be devoted to a wide range of different medical conditions and developing dictionaries for each condition (including laymen terms) would be very labor-intensive. Additionally, there are many different ways of expressing the same information and the language use in the forum evolves over time. In this paper, we present an alternative: a corpus-driven spelling correction approach. Our method is designed to be conservative and to focus on precision to mitigate one of the major challenges of correcting errors in domain-specific data: the loss of information due to the erroneous correction of already correct domain-specific terms. Although dictionary-based retrieval will capture all mistakes, because any word that is not in the dictionary is considered a mistake, thereby attaining a high recall, its precision will be low. This is a result of words that are correct but not present in the dictionary as they will be erroneously marked as mistakes. Many domain-specific terms will fall in this category. In contrast, data-driven methods can capture patterns to recognize these non-mistakes as correct words and thereby improve precision, while recall could go down as these patterns might miss mistakes, for example because they are common. A data-driven detection approach will thus be more precise than dictionary-based retrieval.

In this paper, we address two research questions:

1. To what extent can corpus-driven spelling correction reduce the out-of-vocabulary rate in medical social media text?
2. To what extent can corpus-driven spelling correction improve accuracy of health-related classification tasks with social media text?

Our contributions are: (1) an unsupervised data-driven spelling correction method that works well on specialized domains with many OOV terms without the need for a specialized dictionary (our lexical normalization pipeline is available at: https://github.com/AnneDirkson/LexNorm); and (2) the first corpus for evaluating mistake detection and correction in a medical patient forum (the corpus is available at https://github.com/AnneDirkson/SpellingCorpus).

The rest of the paper is organized as follows. In Section 2, we briefly review related work. In Section 3, we discuss the datasets we employ (Section 3.1) followed by a detailed description of our methodology (Section 3.2). In Section 4, we present our evaluation results, which are discussed further in Section 5. Lastly, in Section 6, we conclude our paper with final insights and an outline of future work.

## 2. Related Work

### 2.1. Challenges in Correcting Spelling Errors in Medical Social Media

A major challenge for correcting spelling errors in small and highly specialized domains is a lack of domain-specific resources. This complicates the automatic creation of relevant dictionaries and language models. Moreover, if the dictionaries or language models are not domain-specific enough, there is a high probability that specialized terms will be incorrectly marked as mistakes. Consequently, essential information may be lost as these terms are often key to knowledge extraction tasks (e.g., a drug name) and to specialized classification tasks (e.g., whether the post contain a side effect of "drug X").

This challenge is further complicated by the dynamic nature of language on medical social media: in both the medical domain and social media novel terms (e.g., a novel drug names) and neologisms (e.g., group-specific slang) are constantly introduced. Unfortunately, professional clinical lexicons are also unsuited for capturing the domain-specific terminology on forums, because laypersons and health care professionals express health-related concepts differently [17]. Another complication is the frequent misspellings of key medical terms, as medical terms are typically difficult to spell [10]. This results in an abundance of common mistakes in key terms, and thus, a large amount of lost information if these terms are not handled correctly.

### 2.2. Lexical Normalization of Social Media

The emergence of social networks and text messaging has redefined spelling correction to the broader task of lexical normalization, which may also involve tasks such as abbreviation expansion [15]. In earlier research, text normalization for social media was mostly unsupervised or semi-supervised (e.g., [18]) due to a lack of annotated data. These methods often pre-selected and ranked correction candidates based on phonetic or lexical string similarity [18,19]. Han et al. [19] additionally used a trigram language model trained a large Twitter corpus to improve correction. Although these methods do not rely on training data to correct mistakes, they do rely on dictionaries to determine whether a word needs to be corrected [18,19]. The opposite is true for modern supervised methods: they do not rely on dictionaries but do rely on training data for both misspelling detection and correction. For instance, the best performing method at the ACL W-NUT shared task of 2015 used canonical forms in the training data to develop their own normalization dictionary [20]. Other competitive systems were also supervised and often used deep learning to detect and correct mistakes [21,22] (for more details on W-NUT systems, see Baldwin et al. [15]). More recent competitive results for this shared task include MoNoise [16]. As mentioned, this system is also supervised and uses a lookup list of all replacement pairs in the training data as an important feature in their spelling correction. Since such specialized resources (appropriate dictionaries or training data) are not available for medical forum data, a method that relies on neither is necessary. We address this gap in this paper.

Additionally, recent approaches (e.g., [5]) often make use of language models for spelling correction. Language models, however, require a large corpus of comparable text from the same genre and domain [5], which is a major obstacle for employing such an approach in niche domains. Since forums are often highly specialized, the resources that could capture the same language use are limited. Nevertheless, if comparable corpora are available, language models can contribute to effectively reducing spelling errors in social media [5] due to their ability to capture the context of words and to handle the dynamic nature of language.

Recent developments in the NLP field towards distributional language models based on byte-pair (BPE) or character-level encoding instead of word-level encoding call into question the need for prior spelling correction. In general, character-level models are more robust to noise than word-level models, as they can exploit the remaining character structure regardless of errors. Niu et al. [23] recently developed a character-level attentional network model for medical concept normalization in social

media which can alleviate the problem of out-of-vocabulary (OOV) terms by using a character-level encoding. Their model is robust to misspellings resulting from double characters, swapping of letters, adding hashtags and deletions.

However, firstly, the robustness to noise of character-based models appears to rely on whether they have been trained on noisy data [24,25]. Otherwise, they are prone to breaking when presented with synthetic or natural noise [24,25]. Thus, if sufficiently large amounts of data with similar types of noise are available, character-based models may negate the need for spelling correction. However, in domains lacking such resources, spelling correction in the pre-processing stage is still needed. Secondly, character-based models have computational disadvantages: their computational complexity is higher and it becomes harder to model long-range dependencies [25]. Alternatively, word embeddings designed to be robust to noise [26] could be used. However, this method also requires sufficiently large amounts of comparable noisy data. To provide an indication, Malykh et al. [26] used the Reuters CV-1 corpus consisting of 800,000 news stories ranging from a few hundred to several thousand words in length [27] to generate their robust English word embeddings.

### 2.3. Lexical Normalization of Clinical Records

Similar to medical social media, clinical notes made by doctors are user-generated and noisy. In fact, Ruch et al. [28] reported about one spelling error per five sentences. However, most normalization research for clinical notes has focused on concept normalization instead of lexical normalization [1]. A prominent shared task for concept normalization of clinical notes is Task 2 of the CLEF e-Health workshop in 2014. Its aim was to expand abbreviations in clinical notes by mapping them to the UMLS database [7]. The best system by Wu et al. [29] applied four different trained tagging methods depending on the frequency and ambiguity of abbreviations. Unfortunately, the abbreviations used by doctors are not the same as the ones used by patients, and thus these methods do not transfer.

To correct misspellings in clinical notes, Lai et al. [8] developed a spell checker based on the noisy channel model by Shannon [30]. Noisy channel models interpret spelling errors as distortions of a signal by noise. The most probable message can then be calculated from the source signal and noise models. This is how spelling correction is modeled traditionally [31]. Although their correction accuracy was high, their method relies on an extensive dictionary compiled from multiple sources to detect mistakes. Similarly, the method by Patrick et al. [9] also uses a compiled dictionary for detecting errors. For correction, Patrick et al. [9] used edit distance-based rules to generate suggestions which were ranked using a trigram model. Fivez et al. [32] was the first to leverage contextual information to correct errors in clinical records. They developed an unsupervised, context-sensitive method that used word and character embeddings to correct spelling errors. Their approach outperformed the method proposed by Lai et al. [8] for the benchmark MIMIC-III [33]. However, they did not perform any mistake detection, as they simply tried to correct the annotated misspellings of MIMIC-III. In conclusion, the methods developed for spelling correction in clinical records either only focus on correction or rely solely on extensive, compiled dictionaries to find mistakes. Therefore, they are not applicable in domains lacking such resources.

## 3. Materials and Methods

### 3.1. Data

#### 3.1.1. Data Collection

For evaluating spelling correction methods, we used an international patient forum for patients with Gastrointestinal Stromal Tumor (GIST). It is moderated by GIST Support International (GSI). This dataset was donated to Dr. Verberne by GSI in 2015. We used a second cancer-related forum to assess generalizability of our methods: a sub-reddit community on cancer, dating from 16 September 2009 until 02 July 2018. (http://www.reddit.com/r/cancer) It was scraped using the Pushshift Reddit

API (https://github.com/pushshift/api). The data were collected by looping over the timestamps in the data. This second forum is roughly four times larger than the first in terms of the number of tokens (see Table 1).

**Table 1.** Raw data without punctuation. IQR, Inter-quartile range.

|  | GIST Forum | Reddit Forum |
|---|---|---|
| # Tokens | 1,255,741 | 4,520,074 |
| # Posts | 36,277 | 274,532 |
| Median post length (IQR) | 20 (35) | 11 (18) |

### 3.1.2. Data Annotation

Spelling mistakes were annotated for 1000 randomly selected posts from the GIST data. Each token was classified as a mistake (1) or not (0) by the first author. For the first 500 posts, a second annotator checked if any of the mistakes were false positives. In total, 99 of the 109 non-word spelling errors were annotated for correction experiments. The remaining 10 errors were found later during error detection experiments and were therefore only included in these experiments. The corrections for the 53 unique mistakes present in the first 500 posts were annotated individually by two annotators, of which one was a GIST patient and forum user. Annotators were provided with the complete post in order to determine the correct word. The initial absolute agreement was 89.0%. If a consensus could not be reached, a third assessor was used to resolve the matter. The remaining mistakes were annotated by the first author. For the correction "reoccurrence", the synonym "recurrence" was also considered correct. As far as we are aware, no other spelling error corpora for this domain are publicly available.

To tune the similarity threshold for the optimal detection of spelling mistakes, we used 60% of the annotated data as a development set. The split was done per post and stratified on whether a post contained mistakes or not. Since the data are extremely unbalanced, we balanced the training data to some extent by combining the mistakes with a ten-fold of random correct words with the same word length distribution (see Table 2). These words were not allowed to be numbers, punctuation or proper nouns, because these are ignored by our error detection process. The development set was split in a stratified manner into 10 folds for cross-validation.

**Table 2.** Annotated data for spelling detection experiments. * Excluding punctuation, numbers and proper nouns.

|  | Mistakes (%) | Total Word Count * |
|---|---|---|
| Training set | 57 (9.1%) | 627 |
| Test set | 45 (0.42%) | 10,760 |

### 3.1.3. Corpus for Calculating Weighted Edit Matrix

Since by default all edits are weighted equally when calculating Levenshtein distance, we needed to compute a weighted edit matrix in order to assign lower costs and thereby higher probabilities to edits that occur more frequently in the real world. We based our weighted edit matrix on a corpus of frequencies for 1-edit spelling errors compiled by Peter Norvig (http://norvig.com/ngrams/count_1edit.txt). This corpus is compiled from four sources: (1) a list of misspellings made by Wikipedia editors; (2) the Birkbeck spelling corpus; (3) the Holbrook corpus; and (4) the Aspell error corpus.

### 3.1.4. Specialized Vocabulary for OOV Estimation in Cancer Forums

To be able to calculate the number of out-of-vocabulary terms in two cancer forums, a specialized vocabulary was created by merging the standard English lexicon CELEX [34] (73,452 tokens), the NCI Dictionary of Cancer Terms [35] (6038 tokens), the generic and commercial drug names from the RxNorm [36] (3837 tokens), the ADR lexicon used by Nikfarjam et al. [37] (30,846 tokens) and our

in-house domain-specific abbreviation expansions (DSAE) (42 tokens) (see Section 3.2.1 for more detail). As many terms overlapped with those in CELEX, the total vocabulary consisted of 118,052 tokens (62.2% CELEX, 5.1% NCI, 26.1% ADR, 6.5% RxNorm and <0.01% DSAE).

### 3.2. Methods

#### 3.2.1. Preprocessing

URLs and email addresses were replaced by the strings -URL- and -EMAIL- using regular expressions. Furthermore, text was lower-cased and tokenized using NLTK. The first modules of the normalization pipeline of Sarker [5] were employed: converting British to American English and normalizing generic abbreviations (see Figure 1). Some forum-specific additions were made: Gleevec (British variant: Glivec) was included in the British–American spelling conversion, one generic abbreviation expansion that clashed with a domain-specific one was substituted (i.e., "temp" defined as *temperature* instead of *temporary*), and two problematic medical terms were removed from the slang dictionary (i.e., "ill" corrected to "i'll" and "chronic" corrected to "marijuana").

Moreover, the abbreviations dictionary by Sarker [5] was lower-cased. As apostrophes in contractions are frequently omitted in social media posts (e.g., "im" instead of "i'm"), we expanded contractions to their full form (e.g., "i am"). Firstly, contractions with apostrophes were expanded and subsequently those without apostrophes were expanded only if they were not real words according to the CELEX dictionary. Lastly, domain-specific abbreviations were expanded with a lexicon of domain-specific abbreviations expansions (DSAE). The abbreviations were manually extracted from 500 randomly selected posts of the GIST forum data. This resulted in 47 unique abbreviations. Two annotators, of which one was a domain expert, individually determined the correct expansion term for each abbreviation, with an absolute agreement of 85.4%. Hereafter, they agreed on the correct form together (This abbreviations lexicon is shared at https://github.com/AnneDirkson/LexNorm).

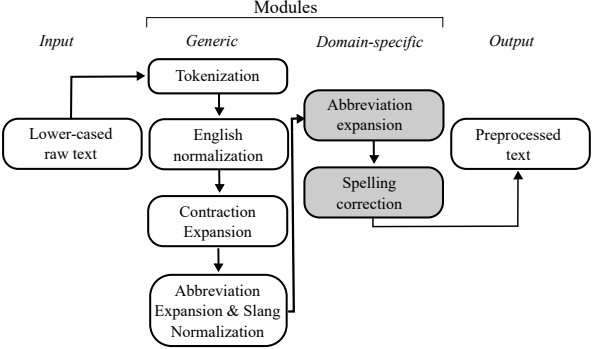

**Figure 1.** Sequential processing pipeline.

#### 3.2.2. Spelling Correction

Baseline Methods

We used the method by Sarker [5] as a baseline for spelling correction. Their method combines normalized absolute Levenshtein distance with Metaphone phonetic similarity and language model similarity. For the latter, distributed word representations (skip-gram word2vec) of three large Twitter datasets were used. In this paper, we used the largest available version of the DIEGO LAB Drug Chatter Corpus (around 1 billion tweets) [38], as it was the only health-related corpus of the three. We also used a purely data-driven spelling correction method for comparison: Text-Induced Spelling Correction (TISC) developed by Reynaert [39]. It compares the anagrams of a token to those in a large corpus of text to correct mistakes. These two methods were compared with simple absolute and relative

Levenshtein distance and weighted versions of both. To evaluate the spelling correction methods, the accuracy (i.e., the percentage of correct corrections) was used. The weights of the edits for weighted Levenshtein distance were computed using the log of the frequencies of the Norvig corpus. We used the log to ensure that a $10\times$ more frequent error does not become $10\times$ as cheap, as this would make infrequent errors too improbable. To make the weights inversely proportional to the frequencies and scale the weights between 0 and 1 with lower weights signifying lower costs for an edit, the following transformation of the log frequencies was used: Weight Edit Distance $= \frac{1}{1+log(frequency)}$.

Correction Candidates

Spelling correction methods were first compared using the terms from a specialized vocabulary for cancer forums (see Section 3.1) as correction candidates. This enabled us to evaluate the methods independently of the vocabulary present in the data. Hereafter, we assessed the impact of using correction candidates from the data itself instead, since our aim was to develop a method that is independent of manually compiled lexicons. Numbers, proper nouns and punctuation were ignored as possible correction candidates.

We inspected whether restricting the pool of eligible correction candidates based on their corpus frequency relative to that of the token aids correction. We used relative corpus frequency thresholds ranging from at least 0 times (no restriction) to 10 times more frequent than the token. The underlying idea was that the correct word would be used more often than the incorrect word and by restricting the candidates we prevented implausible but similar words from hindering correction. This, for instance, prevents mistakes from being corrected by other similar and roughly equally frequent mistakes. A relative, instead of absolute, threshold that depends on the frequency of the mistake enabled us to also correct mistakes even if they occur more commonly (e.g., misspellings of a complex medication name). Candidates were considered in order of frequency. Of the candidates with the highest similarity score, the first was selected.

We tried two different approaches to further improve correction by altering the pool of correction candidates. Firstly, we tested whether prior lemmatization of the spelling errors with or without prior lemmatization of the correction candidates could improve spelling correction. Secondly, we investigated the effect on correction if we were to impose an additional syntactic restriction on the correction candidates, namely only allowing those with the same Part-of-Speech tag at least once in the data or the same surrounding POS tags to the left and right (i.e., the POS context) at least once in the data. McNemar tests were used to test whether the predictions of various correction methods are significantly different. In all follow-up experiments, correction candidates were derived from the respective dataset and constrained by the optimal relative corpus frequency threshold.

Improving the Baseline Method

For the best baseline method with data-driven candidates, we explored whether context could aid the correction further using: (1) language models of the forum itself; or (2) publicly available distributed and sequential language models of health-related social media data. This last category includes the distributed word2vec (dim= 400) and sequential trigram language models developed by Sarker and Gonzalez [38] and the distributed word2vec (dim = 200) HealthVec model developed by Miftahutdinov et al. [40]. The models by Sarker and Gonzalez [38] are based on around 1 billion Twitter posts derived from user timelines where at least 1 medication is mentioned. A smaller version of this language model is used in the current state-of-the-art normalization pipeline for general social media [5] (language models can be obtained from: https://data.mendeley.com/datasets/dwr4xn8kcv/3). The HealthVec model is based on the Health Dataset consisting of around 2.5 million user comments from six web resources: WebMD, Askapatient, patient.info, Dailystrength, drugs.com and product reviews from the Amazon Dataset (available at: http://jmcauley.ucsd.edu/data/amazon). Besides employing these language models, we explored whether adding double Metaphone phonetic

similarity [41] improves correction. Phonetic similarity is a measure for how phonetically similar an error is to the potential correction candidate.

The best baseline method was combined with these measures (i.e., the similarity from a language model or phonetic similarity) in a weighted manner with weights ranging from 0 to 1 with steps of 0.1. The inverse weight was assigned to the baseline similarity measure. For all language models, if the word was not in the vocabulary, then the model similarity was set to 0, essentially rendering the language model irrelevant in these cases. To investigate the impact of adding these contextual measures, Pearson's r was used to calculate the correlation between the correction accuracy and the assigned weight.

### 3.2.3. Correcting Concatenation Errors

If a word is not in the Aspell dictionary (available at: http://Aspell.net/), but is also not a spelling mistake, our method checks if it needs to be split into two words. It is split only if it can be split into two words of at least 3 letters which both occur more in the corpus more frequently than the relative corpus frequency boundary. For each possible split, the frequency of the least frequent word is considered. The most plausible split is the one for which this lower frequency is highest (i.e., the least frequent word occurs the most). Words containing numbers (e.g., "3 months") are the exception: they are split so that the number forms a separate word.

### 3.2.4. Spelling Mistake Detection

We manually constructed a decision process, inspired by the work by Beeksma et al. [42], for detecting spelling mistakes. The optimal relative corpus frequency threshold determined for spelling correction in our earlier experiments is adopted. On top of this threshold, the decision process uses the similarity of the best candidate to the token to identify mistakes. If there is no similar enough correction candidate available, then the word is more likely to be a unique domain-specific term we do not wish to correct than a mistake. The minimum similarity threshold is optimized with a 10-fold cross validation grid search from 0.40 to 0.80 (steps of 0.02). The loss function used to tune the parameters was the $F_{0.5}$ score, which places more weight on precision than the $F_1$ score. We believe it is more important to not alter correct terms than to retrieve incorrect ones. Candidates are considered in order of frequency. Of the candidates with the highest similarity score, the first is selected. The error detection automatically labels numbers, punctuation, proper nouns and words present in the Aspell dictionary as correct. We used the word list 60 version of the Aspell dictionary, as is recommended for spelling correction. To verify that medication names were not being classified as proper nouns and thereby excluded from spelling correction, we checked the part-of-speech tags of the most common medication for GIST patients (gleevec) and two of its common misspellings (gleevic and gleevac). For gleevec, 81.4% of the mentions were classified as nouns (NN). The next two largest categories were adjectives (JJ) (7.2%), plural nouns (NNS) (4.7%) and verbs (VB) (3.9%). The remaining 2.8% were divided over 10 POS-tags (ranging from 0.6% to 0.0005%). Most importantly, none were classified as proper nouns (NNP or NNPS). Similarly, gleevic and gleevac were labeled as nouns (NN) 78.1% and 83.9% of the time and neither was ever labeled as a proper noun. For gleevic, the remaining cases were divided amongst plural nouns (11.4%), adjectives (8.3%) and verbs (2.2%). For gleevac, the remainder was divided between verbs (11.9%) and adjectives (4.2%).

We compared our optimized decision process with and without concatenation error detection (see Section 3.2.3) with error detection using two commonly used dictionaries, CELEX [34] and Aspell, with Microsoft Word and with TISC, another data-driven detection method [39]. Significance was calculated with McNemar tests. Any mistakes overlapping between the training and test set were not included in the evaluation.

### 3.2.5. Impact of the Corpus Size on Detection

To measure the influence of the size of the corpus on spelling mistake detection, we varied the size of the corpus from which correction candidates are derived. The token frequencies of errors and candidates were both calculated using this corpus. Therefore, the frequencies of mistakes and potential corrections would vary and we could estimate for each corpus size how much the error detection in 1000 posts would change. We used Jaccard similarity to measure the overlap between the error predictions of each possible combination of two different corpus sizes.

As our relative corpus frequency threshold is a minimal threshold, bigger corpora and thus larger differences between the token frequency of the error and that of the correct variant would not pose a problem. Consequently, we randomly selected posts to artificially downsize our two cancer forums exponentially. We used sizes ranging from 1000 posts to all forum posts. The 1000 posts for which errors were detected were always included in the corpus. For the GIST forum, we used the 1000 annotated posts.

### 3.2.6. Impact of the Degree of Noisiness of the Data

To investigate the impact of the level of noise in the data on spelling correction and detection, we simulated datasets with varying proportions of misspellings. As our method was designed on data with few errors (<1% in our sample), this helped us to understand to what extent our method can generalize to more noisy user-generated data. We generated artificial data by altering the amount of misspellings in two cancer-related fora.

In line with the work by Niu et al. [23], we generated artificial noise typical of social media text by: (i) deleting a single letter; (ii) doubling a letter; and (iii) swapping two adjacent letters. Niu et al. [23] also added hashtags to words, but as this is only relevant for Twitter we omitted this transformation. Words are randomly selected based on a pre-determined probability of occurrence (1%, 2%, 3%, 4%, 8% and 16%). Which letter is removed or swapped in the word is dependent on the normalized likelihood of a deletion or swap occurring in real-word data. We use the normalized log frequencies of the Norvig corpus [43] (additionally normalized to sum to 1 for all possibilities per word). Which letter is doubled is randomly selected, as frequencies for such operations are not available. We evaluated the spelling correction and detection for each forum with the average of three runs of 1000 randomly selected posts with 3 different seeds.

### 3.2.7. Effect on OOV Rate

The percentage of out-of-vocabulary (OOV) terms is used as an estimation of the quality of the data: less OOV-terms and thus more in-vocabulary (IV) terms is a proxy for cleaner data. As the correction candidates are derived from the data itself, one must note that words that are not part of Aspell may also be transformed from IV to OOV. OOV analysis was done manually.

### 3.2.8. External Validation

To evaluate the impact of lexical normalization as a preprocessing step on the performance of separate downstream tasks, we performed extrinsic evaluation of our pipeline by running six text classification experiments. We obtained six publicly available health-related Twitter datasets ranging in size from 588 to 16,141 posts (see Table 3). As can be seen in Table 3, the datasets also have varying degrees of imbalance. It is not uncommon for social media datasets to be highly imbalanced and thus we investigated whether the impact of spelling correction is influenced by imbalance. The datasets were retrieved from the data repository of Dredze (http://www.cs.jhu.edu/~mdredze/data/) and the shared tasks of Social Media Mining for Health Applications workshop (SMM4H) 2019 (https://healthlanguageprocessing.org/smm4h/challenge/).

Text classification was performed before and after normalization using default sklearn classifiers: Stochastic Gradient Descent (SGD), Multinomial Naive Bayes (MNB) and Linear Support Vector

Machines (SVC). Unigrams were used as features. A 10-fold cross-validation was used to determine the quality of the classifiers and a paired t-test was applied to determine significance of the absolute difference. Only the best performing classifier is reported per dataset. For the shared tasks of the SMM4H workshop, only the training data were used.

**Table 3.** Six classification datasets of health-related Twitter data. * SMM4H, Social Media Mining for Health Applications workshop.

| Dataset | Task | Size | Positive Class |
|---|---|---|---|
| Task 1 SMM4H 2019 * | Presence adverse drug reaction | 16,141 | 8.7% |
| Task 4 SMM4H 2019 * Flu vaccine | Personal health mention of flu vaccination | 6738 | 28.3% |
| Flu Vaccination Tweets [44] | Relevance to topic flu vaccination | 3798 | 26.4% |
| Twitter Health [45] | Relevance to health | 2598 | 40.1% |
| Task4 SMM4H 2019 * Flu infection | Personal health mention of having flu | 1034 | 54.4% |
| Zika Conspiracy Tweets [46] | Contains pseudo-scientific information | 588 | 25.9% |

To evaluate our method on generic social media text, we used the test set of the ACL W-NUT 2015 task [15]. The test set consists of 1967 tweets with 2024 one-to-one, 704 one-to-many, and 10 many-to-one mappings. We did not need to use the training data, as our method is unsupervised. We omitted the expansion of contractions from our normalization pipeline for the W-NUT task, because expanding contractions was not part of the goals of the task. Error analysis was done manually on the 100 most frequent errors.

## 4. Results

In this section, we report the distribution of spelling errors in our corpus (Section 4.1), the evaluation of spelling correction (Section 4.2) and detection methods (Section 4.3) on our spelling corpus, the impact of corpus size (Section 4.4) and the level of noise in the corpus (Section 4.5) on the efficacy of our method. We assessed the impact of our method on the OOV rate in two cancer-related fora (Section 4.6) and on classification accuracy of six health-related Twitter benchmarks (Section 4.7). We also evaluated the performance of our method on the W-NUT shared task for generic social media normalization (Section 4.7).

### 4.1. Error Distribution

Spelling errors can be divided into non-word errors (i.e., errors that are not valid words) and real-word errors (i.e., errors that result in another valid word) [47]. Incorrect concatenations and splits can be either. For example, "scan" to "scant" is a real word error whereas "side effects" to "sideeffects" is a non-word error. We focus on correcting non-word errors, as we are not interesting in correcting syntactic or semantic errors [47].

Nonetheless, we investigated the prevalence of these error types in the data to gain insight into which types of errors are made in medical social media text. As can be seen in Table 4, our corpus of 1000 medical posts from the GIST forum mainly contains non-word errors. Moreover, non-word errors contain the highest percentage of medical misspellings (47.7%). Comparatively, only 20% of real word errors are medical terms. Most posts do not contain any errors (see Figure 2), but, for those that do, there was in most cases only one error per post.

**Table 4.** Error distribution in 1000 GIST posts.

| Error Type | Non-Word | Incorrect Splits | Incorrect Concatenations | Real Word |
|---|---|---|---|---|
| Amount | 109 | 17 | 24 | 30 |
| Non-Medical/Medical | 57/52 | 25/5 | 14/3 | 18/6 |
| Percentage of tokens | 0.32% | 0.05% | 0.07% | 0.09% |
| Example mistake | gleevac | gall bladder | sideeffects | scant |
| Example correction | gleevec | gallbladder | side effects | scan |

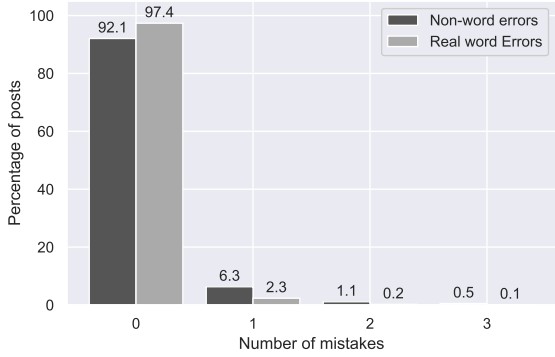

**Figure 2.** Distribution of non-word and real word errors across posts in the GIST forum.

## 4.2. Spelling Correction

The normalization step prior to spelling correction (see Figure 1) corrected 12 of the 99 spelling errors, such as "feelin" to "feeling". These errors are all on the fuzzy boundary between spelling errors and slang. Thus, spelling correction experiments were performed with the remaining annotated 87 spelling errors.

The state-of-the-art method for generic social media by Sarker [5] performed poorly for medical social media: it corrected only 19.3% of the mistakes (see Table 5). In fact, it performed significantly worse ($p < 0.0001$) than all edit distance based methods. Computationally, it was also much slower (see Table 6). A second established data-driven approach, TISC [39], performed even more poorly (14.8%). This was also significantly worse than all edit-based methods ($p < 0.0001$). Relative weighted edit distance performed the best with an accuracy of 68.2%. The theoretical upper bound for accuracy was 92.0%, because not all corrections occur in the specialized dictionary. Examples of corrections can be seen in Table 7.

**Table 5.** Correction accuracy using a specialized vocabulary. AE, absolute edit distance; RE, relative edit distance; WAE, weighted absolute edit distance; WRE, weighted relative edit distance. * Only the best corpus frequency threshold is reported.

| Source of Correction Candidates | Ceiling | AE | RE | WAE | WRE | Sarker | TISC |
|---|---|---|---|---|---|---|---|
| Specialized vocabulary | 92.0% | 58.0% | 64.7% | 63.3% | 68.2% | 19.3% | 14.8% |
| GIST forum text * | 97.6% | **73.9%** | **73.9%** | 70.4% | 72.7% | 44.3% | - |

**Table 6.** Mean computation time over 5 runs.

| AE | RE | WAE | WRE | Sarker |
|---|---|---|---|---|
| 13.36 ms | 14.04 ms | 29.45 ms | 32.00 ms | 904.33 ms |

**Table 7.** Corrections by different methods with candidates from a specialized vocabulary. * Gleevec and Sutent are important medications for GIST patients.

| Mistake | Correction | AE | RE | WAE | WRE | Sarker | TISC |
|---|---|---|---|---|---|---|---|
| gleevac | gleevec * | gleevec | gleevec | gleevec | gleevec | colonic | gleevac |
| stomack | stomach | stomach | stomach | smack | stomach | smack | smack |
| ovari | ovary | ovary | ovary | ovary | ovary | ova | atari |
| sutant | sutent * | mutant | mutant | sutent | sutent | mutant | dunant |
| mestastis | metastasis | miscasts | metastasis | metastasis | metastasis | miscasts | mestastis |

However, when using candidates derived from the data themselves, unweighted absolute and relative edit distance performed the best. Relative edit distance accurately corrected 73.9% of all mistakes at a relative corpus frequency threshold ($\theta$) of 9, while absolute edit distance did so at a $\theta$ of 2–5 (see Table 5 and Figure 3). A $\theta$ of 9 means that candidates are only considered plausible if they occur nine times more frequently than the spelling error. We elected to use relative edit distance, because it is more fine-grained than absolute edit distance, especially for short words. Using data-driven candidates increased the theoretical upper bound from 90.2% to 97.6%. This showcases the limitations of using, even domain-specific, dictionaries for correction.

Nonetheless, simply using all words from the data as possible candidates (i.e., a corpus frequency threshold of 0) for every spelling error resulted in a very low correction accuracy (see Figure 3). However, imposing the restriction that the corpus frequency of a viable correction candidate must be at least double (2×) that of the mistake significantly improved correction ($p < 0.0001$) for all correction methods. In that case, for a mistake occurring 10 times, only words occurring at least 20 times were considered. Thus, the assumption that corrections are more common than mistakes appears to hold true. However, at any threshold, all edit distance based methods still significantly ($p < 0.001$) outperformed the state-of-the-art method [5], in line with previous results (Table 5). Examples of corrections with data-driven candidates are given in Table 8.

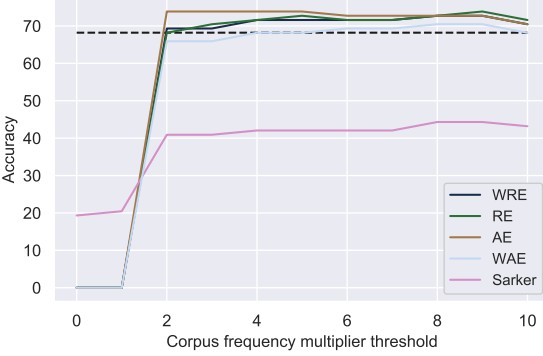

**Figure 3.** Correction accuracy of unique mistakes using correction candidates from the data at various minimum relative corpus frequency thresholds. Dotted line indicates the best correction accuracy using dictionary-derived candidates.

**Table 8.** Corrections by different methods with data-driven candidates. AE, absolute edit distance; RE, relative edit distance; WAE, weighted absolute edit distance; WRE, weighted relative edit distance.

| Mistake | Correction | AE | RE | WAE | WRE | Sarker |
|---|---|---|---|---|---|---|
| gleevac | gleevec | **gleevec** | gleevec | **gleevec** | **gleevec** | **gleevec** |
| stomack | stomach | **stomach** | stomach | **stomach** | **stomach** | stuck |
| ovari | ovary | **ovary** | **ovary** | **ovary** | **ovary** | ovarian |
| sutant | sutent | **sutent** | sutent | sutent | **sutent** | mutant |
| mestastis | metastasis | metastis | metastis | metastis | metastis | metastis |

The accuracy of the best baseline method, namely relative edit distance with a $\theta$ of 9, was unaffected by prior lemmatization of the spelling errors (see Table 9). It thus appears that, if prior lemmatization can correct the error, our method automatically does so. In contrast, additional lemmatization of their corrections and of the correction candidates significantly reduced accuracy ($p = 0.021$ and $p = 0.011$) compared to omitting prior lemmatization. Thus, lemmatization of the data or candidates prior to spelling correction is not recommended.

**Table 9.** Effect of lemmatization of the errors (LemmatizedInput), their corrections (LemmatizedOutput) and correction candidates (LemmatizedCandidates) on spelling correction accuracy using RE ($\theta = 9$).

| NoLemmatization | LemmatizedInput | + LemmatizedOutput | + LemmatizedCandidates |
|:---:|:---:|:---:|:---:|
| 73.6% | 73.6% | 64.7% | 67.0% |

### 4.2.1. Adding Weighted Phonetic Similarity

Previous research has shown that, when users are faced with the task of writing an unfamiliar, complex word such as a drug name, they tend to revert to phonetic spelling [48]. Therefore, we investigated whether adding a weighted phonetic component may improve correction. This is not the case: The weight assigned to phonetic similarity had a strong negative correlation ($-0.92$) with the correction accuracy ($p < 0.0001$) (see Figure 4). This suggests that such phonetic errors are already captured by our frequency-based method.

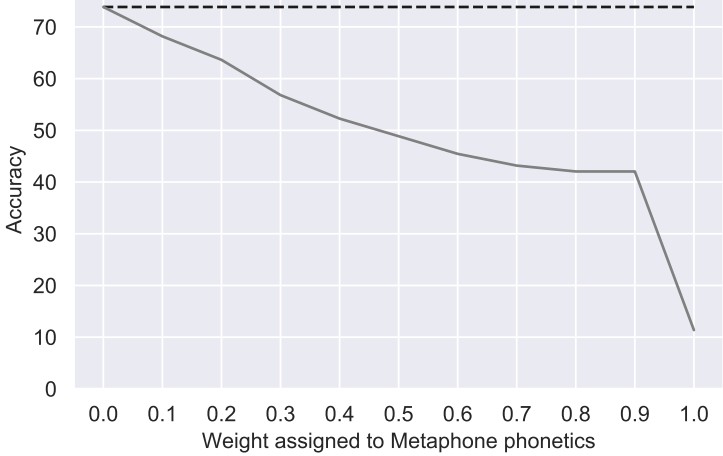

**Figure 4.** Correction accuracy with additional weighted double Metaphone phonetic similarity. Dotted line indicates the best accuracy with relative edit distance alone.

### 4.2.2. Adding Weighted Contextual Similarity

Previous work has indicated that the context of the spelling mistakes might be helpful to improve spelling correction [32]. Since domain-specific resources are scarce, one potential approach is to use the contextual information present in the corpus itself. Based on work by Beeksma et al. [42], we tried to use the Part-of-Speech (POS) tags of the error or the POS tags of its neighbors to constrain correction candidates. However, as can be seen in Figure 5, adding these constraints reduced correction accuracy, although not significantly. Aside from some additional errors, using POS context as a constraint resulted in identical errors as enforcing a similar POS tag for potential correction candidates, regardless of whether NLTK or Spacy was used.

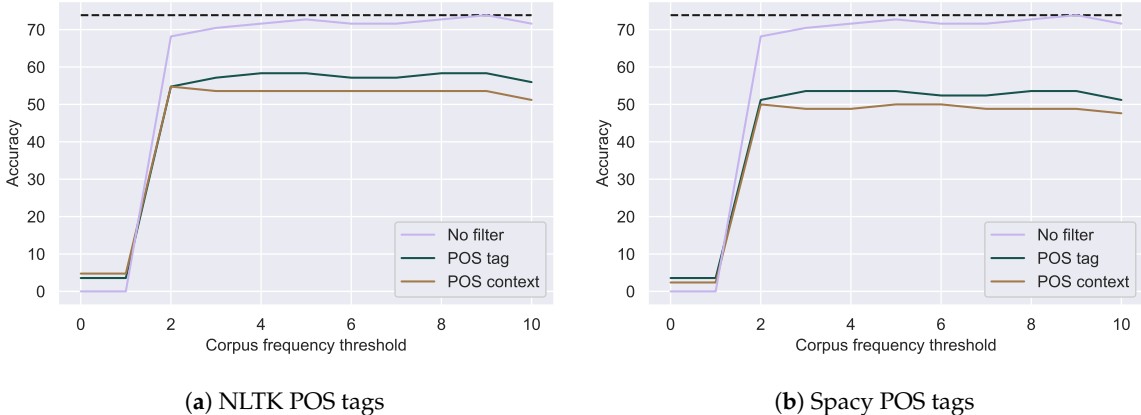

**(a)** NLTK POS tags

**(b)** Spacy POS tags

**Figure 5.** Correction accuracy of spelling mistakes with additional POS tag filters using NLTK (**a**) or Spacy (**b**). Dotted lines indicate the best accuracy with relative edit distance alone.

As many modern methods use language models to aid spelling correction [5], we also examined whether we can leverage contextual information by using language models of the corpus itself to improve correction accuracy. For both Word2vec and FastText distributed models of the data, we found that the higher is the weight assigned to the language model similarity, the more the accuracy drops. This inverse correlation is significant and almost equal to $-1$ for all dimensionalities ($p < 0.000001$) (see Figure 6a,b). Our data are possibly too sparse for either contextual constraints on the correction candidates or employing language model similarity in this manner. They are also too small for building a sequential trigram model [49].

Alternatively, we could employ more generic language models based on medical social media, but not specific to a particular disease domain. We found that a distributed language model based on a collection of health-related tweets, the DIEGO Drug chatter corpus [38], does not manage to improve correction accuracy (see Figure 6c). Nevertheless, a sequential trigram model based on this same Twitter corpus did improve correction accuracy by 2.2–76.1% at a weight of 0.6 (see Figure 6c). The weight assigned to the probability of a trigram with the correction in place of the error was positively correlated ($r = 0.58$) with the correction accuracy. However, the HealthVec distributed language model could improve the correction accuracy up to 79.5% at a weight of 0.6 (see Figure 6d). Overall, its assigned weight was also positively correlated ($r = 0.63$) with the correction accuracy. Table 10 shows that adding the HealthVec model mostly improved accuracy for non-medical errors (e.g., "explane") and for medical errors for which it is difficult to determine whether they should be singular or plural (e.g., "ovarie" and "surgerys"). One medical term (i.e., "surgerys") was no longer corrected accurately. We opted to employ this weighted method due to its higher overall accuracy, but one could opt to not include the HealthVec model for their corpus depending on the importance of non-medical terms for the downstream task.

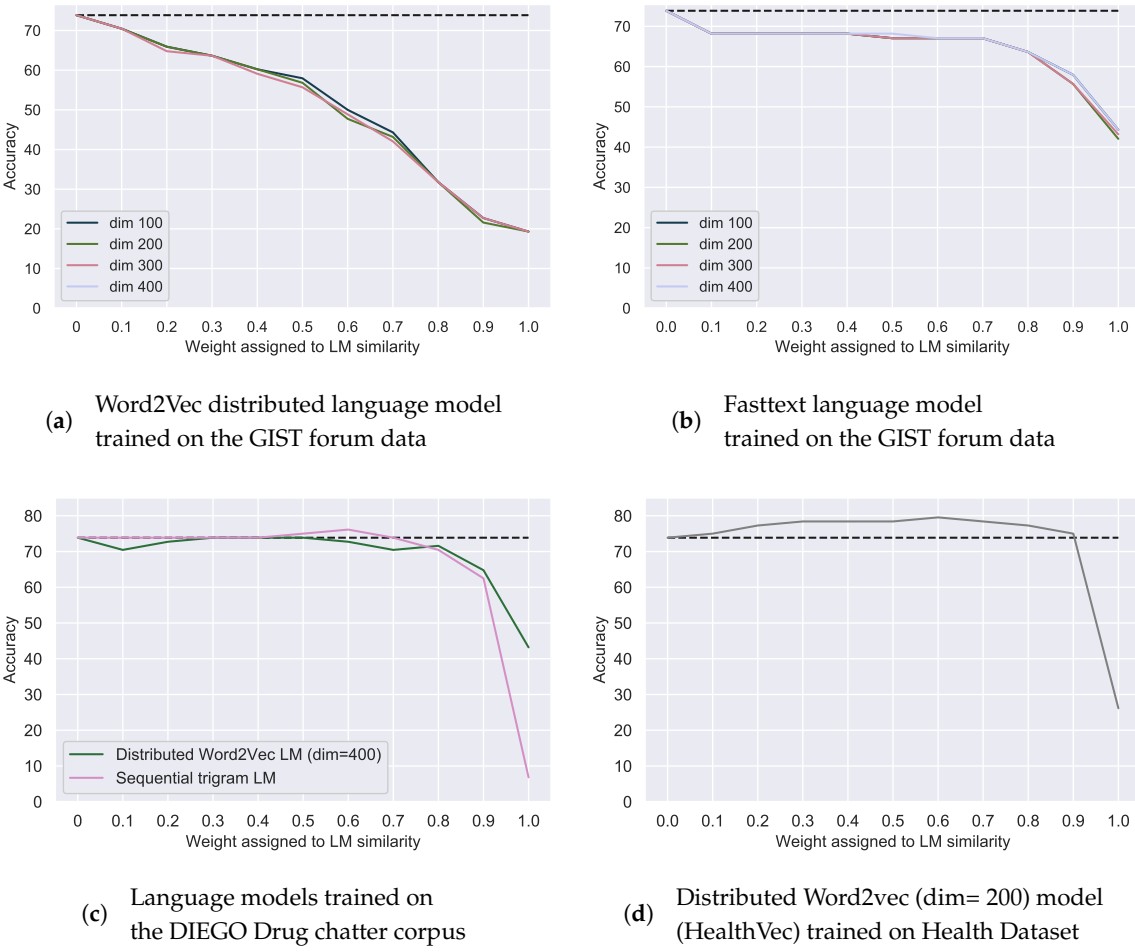

**Figure 6.** Correction accuracy of spelling mistakes with additional weighted language model (LM) similarity using language models of the data itself (**a**,**b**) or trained on external datasets: the DIEGO Drug Chatter corpus (**c**) and the Health Dataset (**d**) . Weight of the LM similarity is the inverse of the weight of the relative edit distance. Dotted line indicates the best accuracy with relative edit distance alone.

**Table 10.** Changes in corrections when HealthVec was added (weight = 0.6) to the relative edit distance (weight = 0.4) with $\theta$ = 9. LM = language model.

| | Error | Correct Word | Correction | |
|---|---|---|---|---|
| | | | **Without LM** | **With LM** |
| Improved | alse | else | false | else |
| | lm | im | am | im |
| | esle | else | resolve | else |
| | explane | explain | explained | explain |
| | ovarie | ovary | ovary | ovaries |
| | surgerys | surgeries | surgeries | surgery |
| Missed | surgerys | surgery | surgery | surgury |

## 4.3. Spelling Mistake Detection

A grid search resulted in an optimal similarity score threshold of 0.76. As higher similarity scores indicate tokens are more dissimilar, this means that, if the best correction candidate has a higher similarity score than this threshold, the token is not corrected (see Figure 7). This combination attained

the maximum $F_{0.5}$ score for 8 of 10 folds. For the other two folds, 0.74 was optimal. See Figure 7 for the tuned decision process. On the test set, our method attained a significantly higher precision ($p < 0.0001$) and $F_{0.5}$ score ($p < 0.0001$) than all other detection methods (see Table 11). Our method attained a slightly lower recall than dictionary-based methods, although its recall was very high at 0.91. Adding concatenation correction to our method improved both recall and precision by 0.05 and 0.01, respectively. See Table 12 for some examples of errors made by our decision process and the corrections our method would output.

Although the recall of generic dictionaries was maximal at 1.0, their precision was low (0.11 and 0.26). Both were logical: The high recall was a result of dictionary-based methods classifying all terms not included in the dictionary as mistakes, which would include all non-word errors, whereas the low precision was a result of the misclassification of correct domain-specific terms that are not included in the dictionary. Aspell outperformed CELEX due to its higher coverage of relevant words such as "oncologist", "metastases" and "facebook". Microsoft Word and TISC performed the worst overall: their precision was low but they also had a lower recall than both dictionary-based methods and our method.

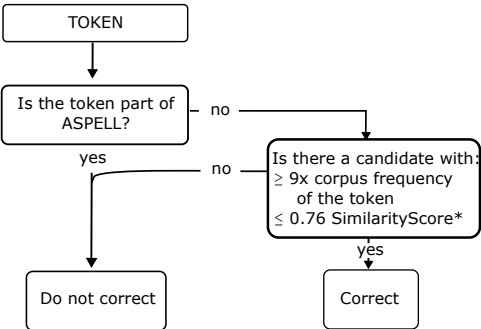

**Figure 7.** Decision process. * SimilarityScore = 0.6 × model similarity + 0.4 × RE.

**Table 11.** Results for mistake detection methods on the test set.

| Method | Mistakes Found | Recall | Precision | $F_{0.5}$ | $F_1$ |
|---|---|---|---|---|---|
| CELEX | 395 | **1.0** | 0.11 | 0.13 | 0.20 |
| Aspell dictionary | 163 | **1.0** | 0.26 | 0.31 | 0.42 |
| TISC | 270 | 0.74 | 0.12 | 0.14 | 0.21 |
| Microsoft word | 395 | 0.88 | 0.10 | 0.12 | 0.18 |
| Our method (RE = 0.76) | 90 | 0.91 | 0.46 | 0.51 | 0.61 |
| Our method (RE = 0.76) + ConcatCorrection | 92 | 0.96 | **0.47** | **0.52** | **0.63** |

**Table 12.** Examples of false positives and negatives of our error detection method.

| Mistakes (their Corrections with our Method) | | | |
|---|---|---|---|
| False positives | intolerances (intolerant) | resected (removed) | reflux (really) | condroma (syndrome) |
| False negatives | istological (histological) | vechile (vehicle) | | |

## 4.4. Impact of Corpus Size

Even though a relative corpus frequency threshold is more robust to different corpus sizes than an absolute one, it is likely that the ratio between tokens and their corrections would vary if the corpus size becomes smaller. Thus, we investigated to what extent the multiplication factor of nine would be robust to such ratio changes.

Figure 8 shows that our threshold was highly robust to corpus size with maximal Jaccard similarity (1.0) for all comparisons. Figure 9 demonstrates this with an example of one common ("gleevac") and one uncommon misspelling ("gllevec") for the medication Gleevec. Each of them has, given its

relative corpus frequency, a respective minimum corpus frequency for each corpus size (indicated by the dotted lines with the same color. Irrespective of the corpus size, the correct variant "gleevec" (the purple line) remains above the minimum corpus frequency (i.e., the dotted lines), for the complete range of corpus sizes.

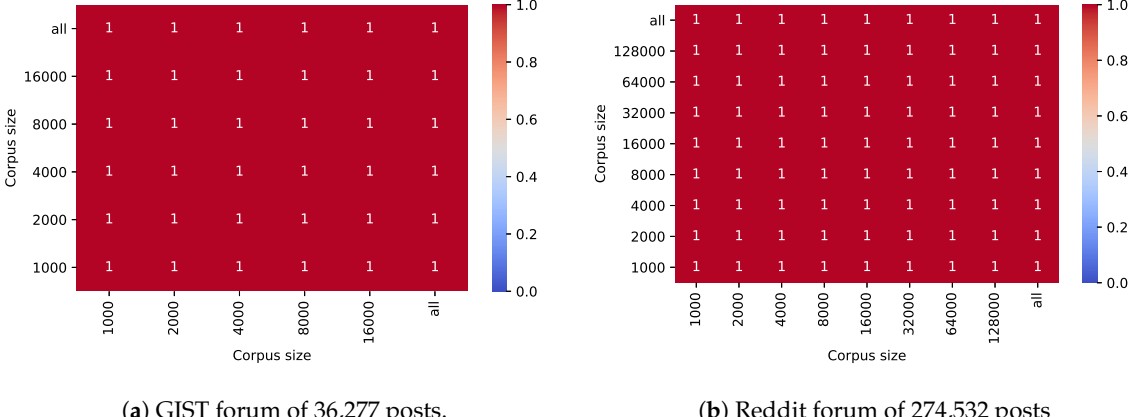

(**a**) GIST forum of 36,277 posts.　　　　　　　　(**b**) Reddit forum of 274,532 posts

**Figure 8.** Stability of error detection in 1000 posts with varying corpus size for the GIST forum (**a**) and the Reddit forum on cancer (**b**).

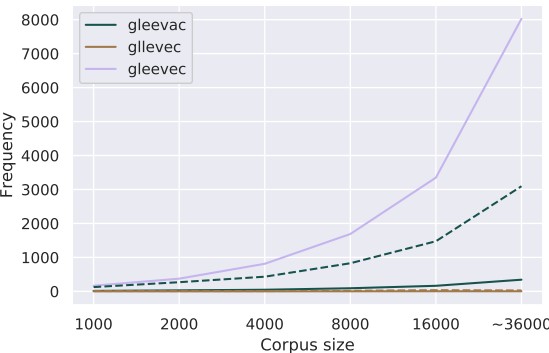

**Figure 9.** Corpus frequency of one uncommon and one common misspelling of the medication Gleevec in the GIST forum with increasing corpus size. The dotted line indicates the corpus frequency threshold for correction candidates for each misspelling.

## 4.5. Impact of the Degree of Noisiness of the Data

As our method was designed on data with few errors (<1% in our sample), we investigated to what extent our method can generalize to more noisy user-generated data using simulated datasets with varying proportions of misspellings. As can be seen in Figure 10a,b, correction accuracy was either stable or increased when the level of noise increased from 1% to 8%, whereas it appeared to diminish at a noise level of 16%. As relative Levenshtein distance does not depend on the noise in the corpus, this possibly indicates that at 16% noise the corpus was affected to the degree that the frequency of correct counterparts of errors often drops below the $\theta$ of nine times the frequency of the error. This is not surprising: due to the equal probability that each word has of being transformed into a mistake, increasingly more words necessary for correction were transformed into errors. However, no conclusions can be drawn about the exact turning point, as we did not measure the impact of noise levels between 8% and 16%. If necessary, re-tuning of the threshold on a more noisy corpus may resolve this issue.

With the exception of errors due to doubling of letters, the absolute correction accuracy was far lower than on our real-world dataset (79.5%). We believe there may be two reasons for this: firstly,

in reality, users are more likely to misspell medical terms than other words [10] and thus this random distribution is unrealistic. Such medical terms are likely to be longer than the average word in social media text. Indeed, we found that in our real-world sample of 1000 posts from the GIST forum the 109 non-word errors are significantly longer ($p < 1^{-22}$) according to a Mann–Whitney U test than the average word: The errors have a mean character length of 6.8 compared to an overall average of 4.2 characters. Since deletions or swaps in shorter words lead to more ambiguous words (e.g., "the" to "te") or even other real words (e.g., "the" to "he"), this would lower the overall correction accuracy of methods designed to correct non-word medical errors. The second reasons ties into this: these artificial datasets do not allow for differentiation between real word and non-word errors and thus are not suited to evaluating absolute non-word error correction. Nonetheless, although absolute accuracy on synthetic data may thus not be reliable indicator, the relative accuracy at different noise levels does provide a good indication for the impact of the level of noise in the data on the efficacy of our method.

Regarding the detection of errors, recall appeared to drop as the level of noise increased for swaps and deletions and remained roughly constant for errors due to doubling of characters (i.e., doubles) (see Figure 10c,d). In contrast, precision increased with increasing noise for swaps and doubles and remained mostly stable for deletions (see Figure 10e,f). These results may indicate that the relative frequency ratios of false positives to their predicted corrections are more frequently close to the detection threshold ($\theta$) of nine than those of true positives. As an artificial increase in noise by a certain percentage (e.g., 4%) would cause the frequency of correct words to drop by approximately that percentage due to random chance selection of words to transform into errors, increasing noise would lead to a slight drop in the ratio between a token and its predicted correction. If the ratio were far larger than nine, it would not alter the outcome. However, if the ratio were only slightly above nine, then it might drop below the detection threshold when the noise was increased. In that case, the token would no longer be marked as an error. Thus, if false positives more frequently have ratios slightly above nine than true positives do, this could explain the increase in precision.

To investigate this idea, we considered swaps in the GIST forum at different levels of noise. It appeared that indeed false positives had a higher percent of ratios between 9 and 10 than true mistakes at lower noise levels (2%, 4% and 8%) across all random seeds. This flips for 16%: false positives now had a lower percentage of ratios liable to drop below the $\theta$ of nine than true positives. Thus, possibly, the false positives that were "at risk" for dropping below the required $\theta$ had done so. This increased precision did come at a cost: some errors also had ratios close to nine, leading to a drop in recall with increasing noise levels.

Due to the presence of common errors, the impact of noise might be less pronounced for real data. Although the artificial data do contain common errors (e.g., "wtih" (218×)), their frequency depends on the frequency of the word of origin (e.g., "with" (9635×)) because each word has an equal, random probability of being altered. Consequently, their ratio would be much higher and they would be easier to detect than real common errors. Moreover, absolute precision and recall on synthetic data may not be transferable. Overall relative trends, however, do provide a first indication for the generalizability of our method to noisier datasets. Further experimentation with noisier, annotated real world data will be necessary to assess the true effect of noise on our error detection.

For both error correction and detection, results are consistent across the two forums and variance of the results is low except at tail end (16%). This can be explained by the random assignment of transformations for each run: depending on which words are randomly transformed in a certain run, the frequency of certain correct words may either fall below the $\theta$ of 9 or not.

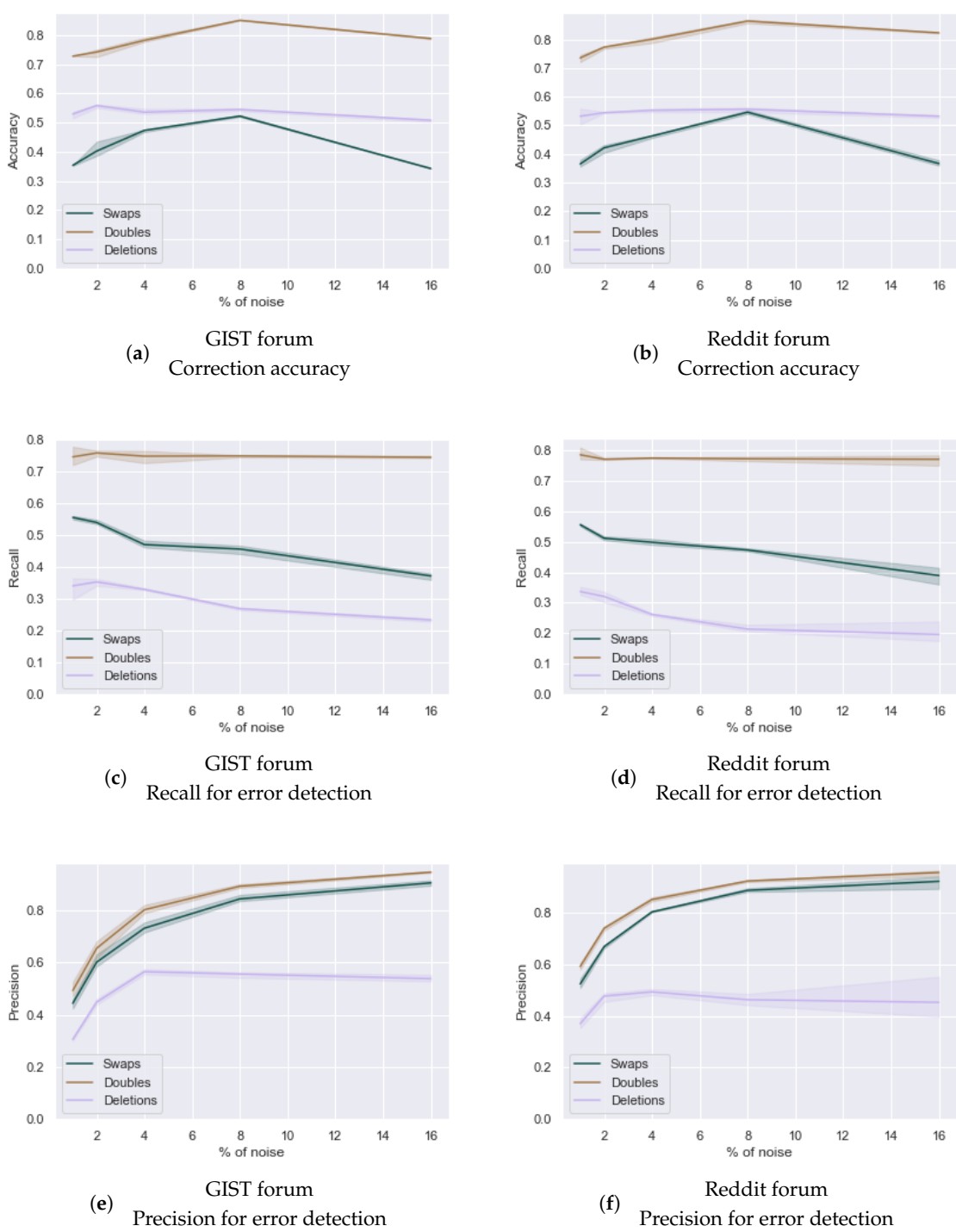

**Figure 10.** Impact of degree of noisiness of the data (1%, 2%, 4%, 8% and 16% noise) on the detection (**c**–**f**) and correction accuracy (**a,b**) of three types of spelling errors (deletions of a single letter, doubling of a single letter and swaps of adjacent letters) in two cancer-related forums. The lines indicate the mean result while the band indicates the variance in results over three runs.

## 4.6. Effect on OOV Rate

The reduction in out-of-vocabulary (OOV) terms was higher for the GIST (0.64%) than for the Reddit forum (0.36%) (see Figure 11b). As expected, it appeared that in-vocabulary terms were

occasionally replaced with out-of-vocabulary terms, as the percentage of altered words was higher than the reduction in OOV (0.72% vs. 0.64% for the GIST and 0.50% vs. 0.36% for the Reddit forum). The vast majority of the posts do not contain any mistakes and, of the posts with mistakes, the majority have only one (see Figure 11a). Thus, it appears that the spelling mistakes were not caused by a select group of individuals who are poor at spelling, but by various forum users making the occasional mistake.

Interestingly, the prior OOV count of the GIST forum is more than double that of the sub-reddit on cancer. This could be explained by the more specific nature of the forum: it may contain more words that are excluded from the dictionary, despite the fact that the dictionary is tailored to the cancer domain. This again underscores the limitations of dictionary-based methods.

Many of the most frequent corrections made in the GIST forum are medical terms (e.g., gleevec, oncologists, and tumors). Similarly, the most frequent mistakes found in this forum are common misspellings of medical terms (e.g., gleevac and gleevic) (see Figure 12a). It appears that, for common medical corrections, there are often various less commonly occurring misspellings per term since their misspelt equivalents do not show up among the most common mistakes. We also found that our method normalizes variants of medical terms to the more prevalent one (e.g., reoccurrence to recurrence). Thus, although the overall reduction in OOV-terms may seem minor, our approach appears to target medical concepts, which are highly relevant for knowledge extraction tasks. In addition, our method incorrectly alters plural to singular variants (e.g., gists to gist), probably due to their higher prevalence in the data. Additionally, due to the addition of the distributed language model, prevalent terms can be replaced by their synonyms as "corrections" (e.g., resected to removed). Fortunately, the resulting information loss would be minimal for medical downstream tasks.

In the sub-reddit on cancer, frequent corrections include medical terms (e.g., chemotherapy, medication and hospital), normalization from plural to singular (e.g., "wifes" to "wife") but also both incorrect alterations of slang (e.g., "gon" to "got") and of medical terms (e.g., immunotherapy) (see Figure 12b). Additionally, the change from "didn" to "did" is problematic due to the loss of the negation. Our method thus appears to work less well for more general fora.

Nonetheless, when we considered the 50 most frequent remaining OOV terms, only a small proportion of them are in fact non-word spelling errors, although slang words could arguably also be part of this category (see Table 13 for examples). A significant portion consists of real words not present in the specialized dictionary. Importantly, also some drug names and medical slang (e.g., "scanxiety" or "anxiety about being scanned") are considered OOV. Since they can be essential for downstream tasks, it is promising that they have not been altered by our method.

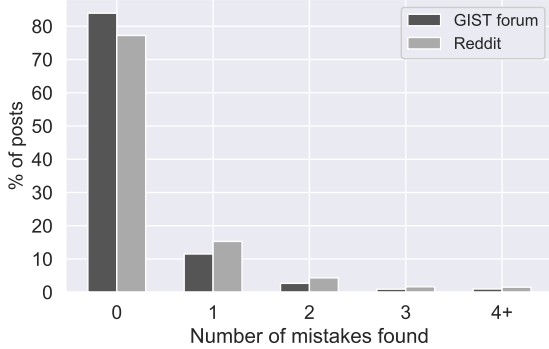

(**a**) Distribution of found mistakes across posts

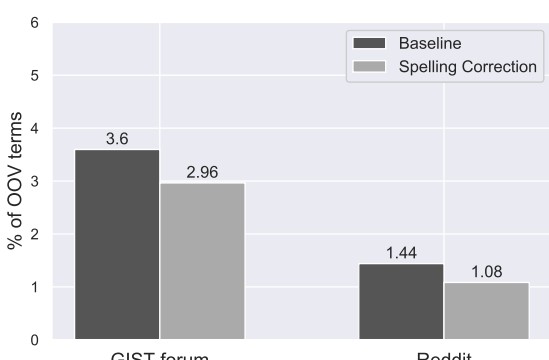

(**b**) Change in out-of-vocabulary terms

**Figure 11.** The impact of spelling correction on the percentage of out-of-vocabulary terms in two cancer forums (**b**) and the distribution of the spelling mistakes that were found (**a**).

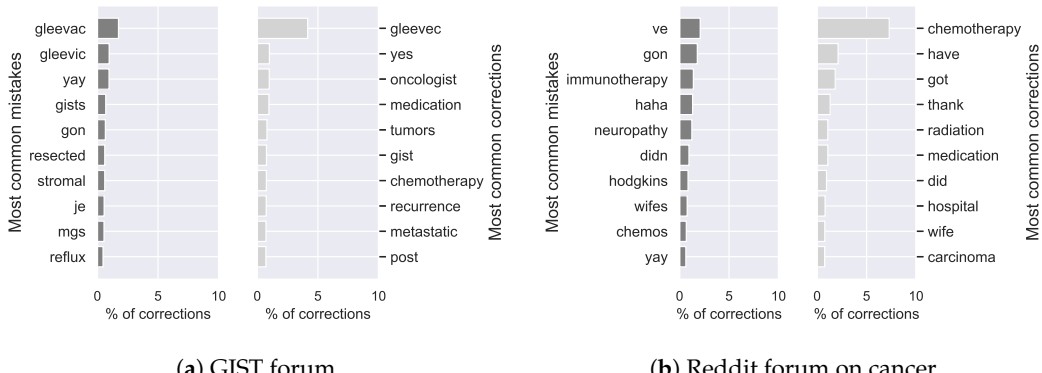

**Figure 12.** Most frequent mistakes and corrections on the GIST forum (**a**) and the Reddit forum on cancer (**b**).

**Table 13.** Manual error analysis of 50 most frequent OOV terms after spelling detection.

|  | GIST | Example | Reddit | Example |
|---|---|---|---|---|
| Real word | 33 | unpredictable, internet | 42 | misdiagnosed, website |
| Spelling mistake | 5 | side-effects, wildtype, copay, listserve, listserve | 2 | side-effects, inpatient |
| Abbreviation | 2 | mos, wk | 3 | aka |
| Slang | 6 | scanxiety, gister | 1 | rad |
| Drug name | 2 | stivarga, mastinib | 1 | ativan |
| Not English | 2 | que, moi | - |  |
| TOTAL | 50 |  | 50 |  |

### 4.7. External Validation

As can been seen in Table 14, normalization led to a significant change in the $F_1$ score for two of the six classification tasks ($p = 0.0096$ and $p = 0.0044$). For the Twitter Health corpus, this change is mostly likely driven by a significant increase in recall ($p = 0.0040$), whereas for the detection of flu infection tweets (Task4 SMM4H2019) it is the precision that was increased significantly ($p = 0.0064$). In general, these changes are of the same order of magnitude as those made by the normalization pipeline of Sarker [5]. Although the overall classification accuracy on Task 1 of the SMM4H workshop was low, this is in line with the low $F_1$ score (0.522) of the best performing system on a comparable task in 2018 [13].

**Table 14.** Mean classification accuracy before and after normalization for six health-related classification tasks. Only the results for the best performing classifier per dataset are reported. ∗∗ indicates $p < 0.005$; ∗ indicates $p < 0.01$; † indicates absolute change.

|  |  | F1 | | | Recall | | | Precision | | |
|---|---|---|---|---|---|---|---|---|---|---|
| Dataset | Words Altered | Pre | Post | Δ † | Pre | Post | Δ † | Pre | Post | Δ † |
| Task1 SMM4H 2019 | 1.53% | 0.410 | 0.410 | −0.0007 | 0.373 | 0.387 | +0.014 | 0.470 | 0.445 | −0.025 |
| Task4 SMM4H 2019 Flu Vaccination | 0.50% | 0.780 | 0.787 | +0.006 | 0.834 | 0.843 | +0.008 | 0.733 | 0.738 | +0.005 |
| Flu Vaccination Tweets | 0.50% | 0.939 | 0.941 | +0.002 | 0.935 | 0.939 | +0.004 | 0.943 | 0.943 | +0.0004 |
| Twitter Health | 0.71% | 0.702 | 0.718 | +0.016 * | 0.657 | 0.685 | +0.028 * | 0.756 | 0.755 | −0.0009 |
| Task4 SMM4H 2019 Flu Infection | 0.57% | 0.784 | 0.800 | +0.012 ** | 0.842 | 0.854 | +0.013 | 0.735 | 0.754 | +0.019** |
| Zika Conspiracy | 0.36% | 0.822 | 0.817 | −0.005 | 0.817 | 0.829 | +0.012 | 0.835 | 0.814 | −0.021 |

Especially the expansion of contractions and the splitting of hash tags (e.g., "#flushot" to '#flu shot") appeared to impact the classification outcome. In contrast, the goal of the task, the relative amount of corrections and the initial result did not seem to correlate with the change in $F_1$ score. The lack of a correlation between the amount of alterations and the change in $F_1$ score may be explained by the weak reliance of classification tasks on individual terms. Unlike in [5], the improvements also

did not seem to increase with the size of the data. This is logical, as we did not rely on training data. The imbalance of the data may be associated with the change in accuracy to some extent: the two most balanced datasets showed the largest increase (see Table 3). Further experiments would be necessary to elucidate if this is truly the case.

On generic social media text, our method performed only slightly worse than the state-of-the-art methods (see Table 15). We did not need to use the training data, as our method is unsupervised. For comparison, our method attained a $F_1$ of 0.726, a precision of 0.728 and a recall of 0.726 on the W-NUT training data.

Error analysis revealed that 46 of the 100 most frequent remaining errors are words that should not have been altered according to the W-NUT annotation (see Table 16). However, in fact, these words are often slang that our method expanded correctly (e.g., info to information). It is thus debatable whether these are errors. Of the remainder, 33 are either uncorrected abbreviations or slang terms. This may partially be explained by the fact that the slang usage of medical forum users differs from the general Twitter population. Lastly, 16 of these 100 can be considered non-word errors that were missed by our method and another four are errors that were correctly detected but corrected inaccurately.

**Table 15.** Results for unconstrained systems of ACL W-NUT 2015.

|  | $F_1$ | Precision | Recall |
|---|---|---|---|
| MoNoise [16] | **0.864** | **0.934** | 0.803 |
| Sarker's method [5] | 0.836 | 0.880 | 0.796 |
| IHS_RD [50] | 0.827 | 0.847 | **0.808** |
| USZEGED [51] | 0.805 | 0.861 | 0.756 |
| BEKLI [52] | 0.757 | 0.774 | 0.742 |
| LYSGROUP [53] | 0.531 | 0.459 | 0.630 |
| Our method | 0.743 | 0.734 | 0.753 |

**Table 16.** Manual analysis of 100 most frequent errors in W-NUT. * also considered non-word mistakes.

| Type of Error | Frequency | Example | Our Correction | W-NUT Annotation |
|---|---|---|---|---|
| Should not have been altered | 46 | info, kinda | information, kind of | info, kinda |
| Abbreviation not or incorrectly expanded | 19 | smh | smh | shaking my head |
| Uncorrected slang | 14 | esp | esp | especially |
| Missed concatenation error * | 6 | incase | incase | in case |
| Missed apostrophe * | 5 | youre | youre | you're |
| Wrong correction | 4 | u | your | your |
| Missed split mistake * | 3 | i g g y | i g g y | iggy |
| Missed non-word spelling mistake | 2 | limites | limites | limits |
| American English | 1 | realise | realize | realise |
| TOTAL | 100 |  |  |  |

## 5. Discussion

The state-of-the-art normalization method for generic social media [5] performed poorly for medical social media with a spelling correction accuracy of only 19.3%. It was significantly outperformed by all edit-based methods regardless of whether the correction candidates were derived from a specialized vocabulary or the data themselves. The highest correction accuracy (73.9%) was attained by unweighted relative edit distance combined with the constraint that correction candidates need to be at least nine times more frequent than the error. This accuracy was further increased by 5.6–79.5% when it was combined with model similarity based on the HealthVec language model. Our preceding decision process could identify mistakes for subsequent correction, with a $F_{0.5}$ of 0.52 and a significantly higher precision than all other methods while retaining a very high recall of 0.96. Additionally, it was almost completely independent of the size of the corpus for two cancer-related

forums, which is promising for its usage in other even smaller, domain-specific datasets. Our method could also function well for more noisy corpora up to a noise level of 8% (i.e., 1 error in every 12.5 words).

In the two cancer forums that we used for evaluation, the spelling correction reduced OOV-terms by 0.64% and 0.36%. Although the reduction may seem minor, relevant medical terms appear to be targeted and, additionally, many of the remaining OOV-terms are not spelling errors but rather real words, slang, names, and abbreviations. Furthermore, our method was designed to be conservative and to focus on precision to mitigate one of the major challenges of correcting errors in domain-specific data: the loss of information due to the "correction" of correct domain-specific terms.

Our method also significantly improved classification accuracy on two tasks, although the absolute change was marginal. On the one hand, this could be because classification tasks do not rely strongly on individual terms. On the other hand, it may be explained by our use of only unigrams as features. Feature extraction would likely also benefit from normalization and could possibly increase performance differences. Further experimentation is required to further assess the full effect of lexical normalization in downstream tasks.

As named entity recognition (NER) tasks rely more strongly on individual terms, we speculate that our method will have a larger impact on such tasks. Unfortunately, NER benchmarks for health-related social media are limited. We investigated three relevant NER tasks that were publicly available: CADEC [11], ADRMiner [37], and the ADR extraction task of the SMM4H 2019. For all three tasks, extracted concepts could be matched exactly to the forum posts, thus negating the potential benefit of normalization. The exact matching can perhaps be explained by the fact that data collection and extraction from noisy text sources such as social media typically rely on keyword-based searching [54].

Our study has a number of limitations. Firstly, the use of OOV-terms as a proxy for quality of the data relies heavily on the vocabulary that is chosen and, moreover, does not allow for differentiation between correct and incorrect substitutions. Secondly, our method is currently targeted specifically at correcting non-word errors and is therefore unable to correct real word errors. Thirdly, the evaluation dataset for developing our method is small: a larger evaluation dataset would allow for more rigorous testing. Nonetheless, as far as we are aware, our corpora are the first for evaluating mistake detection and correction in a medical patient forum. We welcome comparable datasets sourced from various patient communities for further refinement and testing of our method.

## 6. Conclusions and Future Work

To what extent can corpus-driven spelling correction reduce the out-of-vocabulary rate in medical social media text? Our corpus-driven spelling correction reduces the OOV rate by 0.64% and 0.36% in the two cancer-related medical forums we used for evaluation. More importantly, relevant medical terms appear to be targeted.

To what extent can corpus-driven spelling correction improve accuracy of health-related classification tasks with social media text? Our corpus-driven method could significantly improve classification accuracy on two of six tasks. This is driven by a significant increase in precision for one and by a significant increase in recall for the second.

In conclusion, our data-driven, unsupervised spelling correction method can improve the quality of text data from medical forum posts. We demonstrated the success of our method on data from two cancer-related forums. The automatic spelling corrections significantly improve the $F_1$ score for two of six external classification tasks that involve medical social media data. Our method can also be useful for user-generated content in other highly specific and noisy domains, which contain many OOV terms compared to available dictionaries. Future work will include extending the pipeline with modules for named entity recognition, automated relation annotation and concept normalization. Another possible avenue for future work could be to determine whether a word is or is not from the domain at hand (the medical domain in our case) prior to normalization and apply different normalization techniques in either case. Furthermore, despite a lack of domain-specific, noisy corpora for training

character-level language models, it would be interesting to investigate to what extent our spelling correction can improve classification accuracy using character-level language models pre-trained on other source domains.

**Author Contributions:** Methodology, A.D., S.V. and W.K.; software, A.D.; validation, A.D.; investigation, A.D.; data curation, A.D.; writing—original draft preparation, A.D.; writing—review and editing, S.V., A.S. and W.K.; visualization, A.D.; supervision, S.V. and W.K.; and funding acquisition, W.K. and S.V.

**Funding:** This research was funded by the SIDN fonds (https://www.sidnfonds.nl/).

**Acknowledgments:** We would like to thank Gerard van Oortmerssen for his help annotating the data. This research project is part of the Leiden University Data Science Research Programme.

**Conflicts of Interest:** The authors declare no conflict of interest. The funders had no role in the design of the study; in the collection, analyses, or interpretation of data; in the writing of the manuscript, or in the decision to publish the results.

## Abbreviations

The following abbreviations are used in this manuscript:

| | |
|---|---|
| GIST | Gastro-Intestinal Stromal Tumor |
| SMM4H | Social Media Mining for Health Applications |
| UMLS | Unified Medical Language System |
| MIMIC-III | Medical Information Mart for Intensive Care-III |

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
