# Peer review of "Data-Driven Lexical Normalization for Medical Social Media†"

_mti, doi:10.3390/mti3030060_

Round 1

Reviewer 1 Report

This paper presents a data-driven lexical normalization pipeline for medical social media. For experiments, the authors proposed a novel corpus for evaluating mistake detection and correction. For spelling mistake detection, a decision process was constructed manually, inspired by the work by Beeksma et al. The authors adopted baseline classifiers such as Stochastic Gradient Descent (SGD), Multinomial Naive Bayes (MNB) and Linear Support Vector Machine (SVC). Recent advances in neural architectures have made it into a method of choice for modern natural language processing. However, the authors did not use any neural architectures as strong classification models.

Authors made a very compelling work marking up the social media posts. Although due to laboriousness of the process the output corpora could not be of significant size. There was an alternative approach presented in the literature: to introduce misspellings into the existing data (in an artificial way). First of all, a paper of Niu [1] should be mentioned. These authors work in the same domain of medical terms in social media. Besides, we could mention a few recent works on that topic from other domains: [2, 3, 4, 5].

The mentioned approaches could not replace the presented results in the paper in question, although they could supplement a training set for the systems solving downstream tasks in spelling-rich media.

Other concerns: 

1. The related work of lexical normalization of generic social media (Section 2.2) looks like out of date. The paper about "the most comprehensive benchmark" published in 2015. As for now, there are several datasets of user-generated texts about medications such as CADEC, PsyTAR, SMM4H data. Since these datasets are user-generated, there is a natural level of noisy terms like misspellings, grammar errors, etc. There are several ways to create noise in texts automatically, see [1-5]. It would be interesting to see the analysis of these corpora in terms of the number of noisy medical terms. After that, the authors could use all of these datasets to evaluate the proposed model. 

2. "The current state-of-the-art system for this task is a modular pipeline with a hybrid approach to spelling, developed by Sarker [5].". I concern about this statement. Current NLP systems are robust to noise mostly in the texts due to good word/char representations. For normalization of user-generated texts about drugs, the state-of-the-art (SoTA) models are neural networks based on word embeddings and contextualized language models [6, 7]. I highly recommend cite recent works with neural SoTA models in Section 2.2 and discuss the main differences between your work and there works in this section (how did these papers handle spell correlation? can you apply these architectures to your classification task? which dictionaries did other works use? where did they get the data?).

3. Additional experiments with word representations are required. The authors adopted a word2vec model trained only on a small amount of data (267k Twitter posts). The neural models [6,7] used a word2vec model titled HealthVec trained on 2.5 mln. reviews about drugs and health issues. The HealthVec is publicly available https://github.com/dartrevan/ChemTextMining/tree/master/word2vec.

Would it be better for classification to (i) apply spell checker first and then (ii) use LSTM with the bigger word2vec model HealthVec or the contextualized models such as ELMo and BERT? Is there a need to create novel spelling correction module if current word representations trained on big corpora (word2vec, fasttext, char embeddings, BPE embeddings, ELMo, BERT) could be robust to it? Additional investigation with HealthVec (at least!) could benefit this paper.

This paper has a good research topic. Overall, It would, therefore, be better to add the mentioned references [1-7] and discussion about existing corpora and methods in the related work section as well as comparison of several word representation models in the context of proposed work. The manuscript itself requires major changes before it can be accepted.

[1] Niu J. et al. Multi-task character-level attentional networks for medical concept normalization //Neural Processing Letters. – 2018. – С. 1-18.

[2] Belinkov Y., Bisk Y. Synthetic and natural noise both break neural machine translation //arXiv preprint arXiv:1711.02173. – 2017.

[3] Xie Z. et al. Noising and denoising natural language: Diverse backtranslation for grammar correction //Proceedings of the 2018 Conference of the North American Chapter of the Association for Computational Linguistics: Human Language Technologies, Volume 1 (Long Papers). – 2018. – С. 619-628.

[4] Heigold G., Neumann G., van Genabith J. How Robust Are Character-Based Word Embeddings in Tagging and MT Against Wrod Scramlbing or Randdm Nouse? //arXiv preprint arXiv:1704.04441. – 2017.

[5] Malykh V., Logacheva V., Khakhulin T. Robust Word Vectors: Context-Informed Embeddings for Noisy Texts //Proceedings of the 2018 EMNLP Workshop W-NUT: The 4th Workshop on Noisy User-generated Text. – 2018. – С. 54-63.

[6] Tutubalina E. et al. Medical concept normalization in social media posts with recurrent neural networks //Journal of biomedical informatics. – 2018. – Т. 84. – С. 93-102. https://www.sciencedirect.com/science/article/pii/S1532046418301126

[7] Miftahutdinov Zulfat et al. Deep Neural Models for Medical Concept Normalization in User-Generated Texts // Proceedings of ACL 2019, Student Research Workshop. —  Florence, Italy: Association for Computational Linguistics, 2019.  https://www.researchgate.net/publication/334523885_Deep_Neural_Models_for_Medical_Concept_Normalization_in_User-Generated_Texts

Reviewer 2 Report

The paper presents an approach for text normalisation from social media data related to health. Their approach is a hybrid approach that relies on general-domain normalisation followed by a domain-specific normalisation based on edit distances from a large corpus. The authors also show that text normalisation improves many health informatics tasks such as personal health mention classification. The paper will be a useful addition for health informatics research.

I request the authors to clarify or update the paper, as required, for the following comments:

1) Section 1: Line 24: It is not clear why social media normalisation cannot be handled by existing medical text normalisation techniques. For example, the approach presented in this paper could be applied to clinical notes as well. 

2) Page 2: "...on a language model of generic Twitter data.." : Not sure what 'language model' means here. A clarification would help.

3) Page 2: "A data-driven detection approach will be more precise than dictionary-based retrieval, as dictionary-based methods will classify any domain-specific terms not present in the dictionary as mistakes.": Rule-based approaches (which, in this case, would be dictionary-based retrieval) suffer from high precision, low recall. Data-driven approaches help to improve the recall by discovering patterns from the data that may not be captured in the rules/dictionary. On similar lines, would a data-driven approach have a higher recall (and not necessarily be more precise) if it can cover terms not present in the dictionary?

4) Footnote 2: The URL could be updated to the specific project. ( https://github.com/AnneDirkson/SpellingCorpus ? ) 

5) Page 2 states that "Consequently, hand-made lexicons may get outdated ". Since the approach presented in the paper does not handle changes over time, I feel this is not the focus of the paper and should not be mentioned. 

6) Section 3.1.2: Proper nouns are ignored. Would this skip names of drugs as well? That might leave out a lot of errors that actually occur in social media - because names of drugs may not be spelt correctly by laypeople. Page 5 states that names of drugs were added to the lexicon. The two seem incongruous - if proper names are ignored, drugs need not be added to the lexicon.

7) Section 3.1.5: At the end of this section, a reader is not clear about the purpose of the datasets for 'external validation'. Table 3 has a field called '%positive/negative'. It is not clear why this information is important in the context of this paper. The paper does compare the impact of normalisation on the classifier performance - but that appears much later.

8) It would be useful to have a table that mentions the proportion of errors in medical terms when describing the dataset. This would give an estimate of the extent of the problem being tackled. Most of my suggestions below are linked to this one.

9) one generic abbreviation that clashed with domain-specific abbreviation was replaced ('temp'). Was it the only one? Given that the architecture in the paper performs general-domain correction followed by a domain-specific correction, it would be useful to know if the general-domain module would 'correct' medical terms that would not need fixing.

10) A possible extension is to determine a word as a general-domain word or a medical domain word. That might handle the challenge in (9) above. This could be mentioned as future work.

11) Section 3.2.4: "so as to optimize for detecting errors we can accurately correct" is not clear. This part of the sentence seems too generic.

12) Table 4 shows that 2 of the 4 error categories are not related to medical words. Could the authors clarify the proportions in these cases?

Round 2

Reviewer 1 Report

This paper has improved significantly. The authors did add experiments with additional word embeddings model as well as experiments based on the noisiness of the data.

Minor notes on the confusing text in the manuscript:

-- Paper: For medical social media, concept normalization (i.e., the mapping of tokens 28 to standardized concept IDs in an ontology) has been explored more extensively (e.g. [11]).

How can authors know that this task "explored more extensively" if there is only one paper (not even a survey)... I recommend removing this fragment about concept normalization from Introduction since this is a different task (out of the scope of this paper so the readers could be confused) or adding more links.

-- Paper: However, this system is supervised and thus cannot be employed in domains which lack extensive training data, such as medical social media

How many training data does the MoNoise method need? If the authors did not try to apply this method on social media, the overall statement could not be entirely accurate.

-- Paper: a language model of distributed word representations (word2vec) of generic Twitter data

According to Section 3, the authors used distributed word representations of health-related texts (not only Twitter data). I suggest rewriting this phrase.

-- Paper: Yet, also for this method, large enough amounts of comparable noisy data are necessary

If there is a statement like "large enough amount" in the paper, the authors could give readers some clue about the actual size of the data they need. The readers did not read all citations from related work probably, so at least specify how many data used by [16,25] and the tasks from these papers.

-- Paper: Section 3.2.6. Impact of the degree of noisiness of the data

I suggest putting a link to the results' figure in this section since it is not apparent how to find the figure fastly. Why did precision become higher after adding noise?

I believe this paper should be accepted after fixing minor notes in the text.
